# Safety Aware Reinforcement Learning (SARL)

## Abstract

As reinforcement learning agents become increasingly integrated into complex, real-world environments, designing for safety becomes a critical consideration. We specifically focus on researching scenarios where agents can cause undesired side effects while executing a policy on a primary task. Since one can define multiple tasks for a given environment dynamics, there are two important challenges. First, we need to abstract the concept of safety that applies broadly to that environment independent of the specific task being executed. Second, we need a mechanism for the abstracted notion of safety to modulate the actions of agents executing different policies to minimize their side-effects. In this work, we propose Safety Aware Reinforcement Learning (SARL) – a framework where a virtual safe agent modulates the actions of a main reward-based task agent to minimize side effects. The safe agent learns a task-independent notion of safety for a given environment. The task agent is then trained with a regularization loss given by the distance between the native action probabilities of the two agents. Since the safe agent effectively abstracts a task-independent notion of safety via its action probabilities, it can be ported to modulate multiple policies solving different tasks across different environments without further training. We contrast this with solutions that rely on task-specific regularization metrics and test our framework on the SafeLife Suite, based on Conway's Game of Life, comprising a number of complex tasks in dynamic environments. We show that our solution is able to match the performance of solutions that rely on task-specific side-effect penalties on both the primary and safety objectives while additionally providing the benefit of generalizability and portability.

## 1 Introduction

Reinforcement learning (RL) algorithms have seen great research advances in recent years, both in theory and in their applications to concrete engineering problems. The application of RL algorithms extends to computer games (Mnih et al., 2013; Silver et al., 2017), robotics (Gu et al., 2017) and recently real-world engineering problems, such as microgrid optimization (Liu et al., 2018) and hardware design (Mirhoseini et al., 2020). As RL agents become increasingly prevalent in complex real-world applications, the notion of safety becomes increasingly important. Thus, safety related research in RL has also seen a significant surge in recent years (Zhang et al., 2020; Brown et al., 2020; Mell et al., 2019; Cheng et al.; Rahaman et al.).

### 1.1 Side Effects in Reinforcement Learning Environments

Our work focuses specifically on the problem of side effects, identified as a key topic in the area of safety in AI by Amodei et al. (2016). Here, an agent's actions to perform a task in its environment may cause undesired, and sometimes irreversible, changes in the environment. A major issue with measuring and investigating side effects is that it is challenging to define an appropriate side-effect metric, especially in a general fashion that can apply to many settings. The difficulty of quantifying side effects distinguishes this problem from safe exploration and traditional motion planning approaches that focus primarily on avoiding obstacles or a clearly defined failure state (Amodei et al., 2016; Zhu et al., 2020). As such, when learning a task in an unknown environment with complex dynamics, it is challenging to formulate an appropriate environment framework to jointly encapsulate the primary task and side effect problem.

Previous work on formulating a more precise definition of side effects includes work by Turner et al. (2019) on conservative utility preservation and by Krakovna et al. (2018) on relative reachability. These works investigated more abstract notions of measuring side effects based on an analysis of changes, reversible and irreversible, in the state space itself. While those works have made great progress on advancing towards a greater understanding of side effects, they have generally been limited to simple grid world environments where the RL problem can often be solved in a tabular way and value function estimations are often not prohibitively demanding. Our work focuses on expanding the concept of side effects to more complex environments, generated by the SafeLife suite (Wainwright and Eckersley, 2020), which provides more complex environment dynamics and tasks that cannot be solved in a tabular fashion. Turner et al. (2020) recently extended their approach to environments in the SafeLife suite, suggesting that attainable utility preservation can be used as an alternative to the SafeLife side metric described in Wainwright and Eckersley (2020) and Section 2. The primary differentiating feature of SARL is that it is metric agnostic, for both the reward and side effect measure, making it orthogonal and complimentary to the work by Turner et al. (2020).

In this paper, we make the following contributions which, to the best of our knowledge, are novel additions to the growing field of research in RL safety:

- *SARL*: a flexible, metric agnostic RL framework that can modulate the actions of a trained RL agent to trade off between task performance and a safety objective. We utilize the distance between the action probability distributions of two policies as a form of regularization during training.
- A generalizeable notion of safety that allows us to train a safe agent independent of specific tasks in an environment and port it across multiple complex tasks in that environment.

We provide a description of the SafeLife suite in Section 2, a detailed description of our method in Section 3, our experiments and results for various environments in Section 4 and Section 5 respectively, as well as a discussion in Section 6.

## 2 THE SAFELIFE ENVIRONMENT

The SafeLife suite (Wainwright and Eckersley, 2020) creates complex environments of systems of cellular automata based on a set of rules from Conway's Game of Life (Gardner, 1970) that govern the interactions between, and the state (alive or dead) of, different cells:

- any dead cell with exactly three living neighbors becomes alive;
- any live cell with less than two or more than three neighbors dies (as if by under- or overpopulation); and
- every other cell retains its prior state.

In addition to the basic rules, SafeLife enables the creation of complex, procedurally generated environments through special cells, such as a spawner, that can create new cells and dynamically generated patterns. The agent can generally perform three tasks: *navigation*, *prune* and *append* which are illustrated in Figure 1 taken from Wainwright and Eckersley (2020).

The flexibility of SafeLife enables the creation of still environments, where the cell patterns do not change over time without agent interference, and dynamic environments, where the cell patterns do change over time without agent interference. The dynamic environments create an additional layer of difficulty, as the agent now needs to learn to distinguish between variations in the environment that are triggered by its own actions versus those that are caused by the dynamic rules independent of its actions. As described in Section 4, our experiments focus on the prune and append tasks in still and dynamic environments: *prune-still*, *prune-dynamic*, *append-still*, *append-dynamic*.

### 2.1 SAFELIFE SIDE EFFECT METRIC

There are two separate side effect metrics that we use in this paper: a training-time side effect metric that can easily be calculated at every frame, and a separate end-of-episode side effect metric used to validate overall agent safety. These metrics are identical to the side effect metrics described in Wainwright and Eckersley (2020).

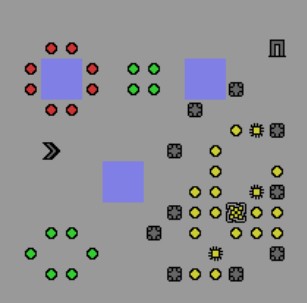

Figure 1: A simple level of the *SafeLife* environment containing an agent (≫), a spawner (▧), crates (▨), and cells of life. The agent's goal is to remove unwanted red cells (*prune task*) and to create new patterns of life in the blue squares (*append task*). Once the agent has satisfactorily completed its goals it can leave via the level exit (⏛). Note that all level boundaries wrap; they have toroidal topology.

The side effect metric used in training is found by comparing any given state to the starting state of that episode. Every cell in the grid that is different from the corresponding cell in the starting state (excluding differences due to agent movement or goal completion) is marked as having a side effect of +1. The total side effect of a state is the sum of side effects of individual cells. The difference in side effects between subsequent states is then to train safe agents as described in Section 3.

At the end of each episode, we perform a separate side effect calculation that is more robust to stochastic and chaotic dynamics observed in SafeLife. We prepare a counterfactual inaction baseline state at time $t_{\mathrm{end}}$, defined as the state that would have been achieved had the agent followed an inaction policy for $t_{\mathrm{end}}$ time steps. We then evolve both the primary state and the inaction state for an additional $n = 1000$ time steps and store the average occupancy of green life cells at each point in the grid, producing two distributions. The total episodic side effect is then given the earth-mover distance between the two distributions using a modified Manhattan distance metric as described in Wainwright and Eckersley (2020). The *normalized* episodic side effect metric, shown in the figures in Section 5, is defined as the total episodic side effect divided by the number of green cells present at the start of the episode, showing the percentage of structures in each episode that are disrupted.

## 3 METHOD

### 3.1 TRAINING FOR REGULARIZED SAFE RL AGENT

Our method relies on regularizing the loss function of the RL agent with the distance of the task agent, $A_\theta$, from the virtual safe actor, $Z_\psi$, as shown in Figure 2.

More formally, the general objective of the task agent $A_\theta$ can be expressed as:

$$\mathcal{F}_\mathcal{A}(\theta) = \mathcal{L}_\theta + \beta * \mathcal{L}_{\mathrm{dist}}(\mathbb{P}_{\pi_\theta}, \mathbb{P}_{\pi_\psi}) \tag{1}$$

where $\beta$ is a regularization hyperparameter, $\mathbb{P}_{\pi_\theta}$ represents the probability of taking a given action given by $A_\theta$, and $\mathbb{P}_{\pi_\psi}$ represents the probability distribution of taking a given action according to $Z_\psi$. As shown in Equation 1, the actor loss $\mathcal{L}_\theta$ is regularized by the distance between the actions suggested by the task agent and the virtual safe agent. The gradient of the objective in Equation 1 expressed as the expectation of rewards of task agent actions $\alpha$ taken from a distribution of policies $\mathbb{P}_{\pi_\theta}$ is then given by:

$$\nabla_\theta \mathcal{F}_\mathcal{A}(\theta) = \nabla_\theta \mathbb{E}_{\alpha \sim \mathbb{P}_{\pi_\theta}}[\mathcal{L}(\theta)] + \beta * \nabla_\theta \mathbb{E}_{\alpha \sim \mathbb{P}_{\pi_\theta}, \zeta \sim \mathbb{P}_{\pi_\psi}}[\mathcal{L}_{\mathrm{dist}}(\alpha, \zeta)] \tag{2}$$

where $\nabla_\theta$ is independent from the virtual safe agent actions $\zeta$ given that $Z_\psi$ is only dependent on $\psi$. This formulation enables training $Z_\psi$ independently from $A_\theta$, thereby abstracting the notion of safety away from the task. The gradient formulation underscores the importance for a distance metric $\mathcal{L}_{\mathrm{dist}}$ that is differentiable to ensure that gradients update the task agent parameters $\theta$ from both terms of the augmented loss functions.

**SARL Agent Training**

Figure 2: A co-training framework for safety aware RL training. The task agent, $A_\theta$, determines the action taken in the environment and the resulting trajectories. The virtual safe agent, $Z_\psi$, receives the same state as $A_\theta$ and makes a suggestion for a safe action given the state. The distance between the action probabilities of the $A_\theta$ and $Z_\psi$ is captured in the Distribution Loss, $\mathcal{L}_{\text{dist}}$. $Z_\psi$ learns how to maximize the safety objective on its own set of environments in parallel to $A_\theta$.

## 3.2 DISTANCE METRICS FOR LOSS REGULARIZATION

The primary objective of the regularization term is to express a notion of distance between a purely reward based action and a purely safety motivated action, thereby penalizing $A_\theta$ for taking a purely reward motivated action. We model the regularization term as the distance between probability distributions $\mathbb{P}_{\pi_\theta}$ and $\mathbb{P}_{\pi_\psi}$.

Given this formulation, previous work (Nowozin et al., 2016; Arjovsky et al., 2017; Huszár, 2015) has provided a number of choices for distance metrics in supervised learning problems with various advantages or shortfalls. One common method of measuring the difference in probability distributions is the KL Divergence, $D_{KL}(p\|q) = \int_x p(x) \log \frac{p(x)}{q(x)} dx$, where $p$ and $q$ are probability distributions described by probability density functions.

The KL Divergence, however, has some significant disadvantages – the most significant one being that the KL Divergence is unbounded when probability density functions to express the underlying distribution cannot be easily described by the model manifold (Arjovsky et al., 2017). Furthermore, the KL divergence is not symmetric given that $D_{KL}(p\|q) \neq D_{KL}(q\|p)$, and also does not satisfy the triangle inequality. One alternative to the KL Divergence is the Jensen-Shannon distance $D_{JS}(p\|q) = \frac{1}{2}D_{KL}(p\|m) + \frac{1}{2}D_{KL}(q\|m)$ with $m = \frac{1}{2}(p+q)$, which is symmetric, satisfies the triangle inequality and is bounded: $0 \leq D_{JS} \leq \log(2)$. These advantages make $D_{JS}$ a good choice for the SARL algorithm, but as discussed extensively in Arjovsky et al. (2017), $D_{JS}$ also has notable disadvantages, the most important being that $D_{JS}$ is not guaranteed to always be continuous and differentiable in low manifold settings.

Another alternative to $D_{JS}$ is the Wasserstein Distance. As discussed in Arjovsky et al. (2017), the Wasserstein Distance is generally better suited for calculating distances for low-dimensional manifolds compared to $D_{JS}$ and other variants of the KL divergence. In its analytical form the Wasserstein Distance $W(p,q) = (\inf_{J \in \mathcal{J}(p,q)} \int \|x - y\|^p \mathrm{d}J(x,y))^{\frac{1}{p}}$, however, is intractable to compute in most cases leading many researchers to establish approximations of the metric. A common way of approximating the Wasserstein Distance is to re-formulate the calculation as an optimal transport problem of moving probability mass from $p$ to $q$, as shown in Cuturi (2013) and Pacchiano et al. (2019). The dual formulation based on behavior embedding maps of policy characteristics described in Pacchiano et al. (2019) is particularly applicable for the SARL algorithm, leading us to adapt it as an additional alternative to the Jensen-Shannon Distance. In this formulation, policy characteristics are converted to distributions in a latent space of behavioral embeddings on which the Wasserstein Distance is then computed.

For our experiments in Section 4, we apply both $D_{JS}$ and the dual formulation of the Wasserstein Distance described in Pacchiano et al. (2019) to compute the distance between $\mathbb{P}_{\pi_\theta}$ and $\mathbb{P}_{\pi_\psi}$.

### 3.3 Safety Aware Reinforcement Learning

The paragraphs above in Section 3 describe the individual components of SARL. The experiments outlined in Section 4 discuss SARL applied to Proximal Policy Optimization (PPO) (Schulman et al., 2017). Wainwright and Eckersley (2020) applied PPO for solving the different environments in the SafeLife suite, making SARL-PPO a natural extension. The loss formulation $\mathcal{L}_\theta^{PPO}$ used in Algorithm 1 is the same as the one decribed in Schulman et al. (2017):

$$\mathcal{L}_\theta^{PPO} = \mathbb{E}_t[\mathcal{L}_t^{Clip}(\theta) - c_1 \mathcal{L}_t^{Value}(\theta) + c_2 S[\pi_\theta](s_t)] \tag{3}$$

As shown in more detail in Algorithm 1, $A_\theta$ is trained using the regularized loss objective described in Equation 1, while $Z_\psi$ is trained exclusively on $\mathcal{L}_\theta^{PPO}$ using the frame-by-frame side effect information as the reward.

---

**Algorithm 1** SARL - PPO

---

 1: Initialize an actor $A_\theta(s)$ and virtual safety agent $Z_\psi(s)$ where $s$ represents the state observation
 2: Initialize a distance metric $\mathcal{L}_{\text{dist}}$
 3: **while** training SARL-PPO **do**
 4:  **for** each actor update $A_\theta$ **do**
 5:   Run $A_\theta(s)$ to generate a minibatch of transitions $\alpha$ with task rewards $r_{\text{task}}$
 6:   Run $Z_\psi(s)$ to generate a minibatch of transitions $\zeta$
 7:   Compute $\mathcal{L}_\theta^{PPO}$ and $\mathbb{P}_{\pi_\theta}$ using transitions in $\alpha$
 8:   Compute $\mathbb{P}_{\pi_\psi}$ using transitions in $\zeta$
 9:   Optimize $A_\theta$ using $\mathcal{L}_\theta^{PPO} + \beta * \mathcal{L}_{\text{dist}}(\mathbb{P}_{\pi_\theta}, \mathbb{P}_{\pi_\psi})$
10:  **end for**
11:  **for** each virtual agent update $Z_\psi$ **do**
12:   Run $Z_\psi(s)$ to generate a minibatch of transitions $\zeta$ with side-effect metric $r_{\text{side}-\text{effect}}$
13:   Compute $\mathcal{L}_\psi^{PPO}$ using transitions in $\zeta$
14:   Optimize $Z_\psi$ using $\mathcal{L}_\psi^{PPO}$ with $r_{\text{side}-\text{effect}}$ as the reward
15:  **end for**
16: **end while**

---

The training algorithm is agnostic to the side effect metric, $r_{\text{side}-\text{effect}}$, used in the environment, leading itself to a plug-and-play approach where the virtual safe agent can modulate the task agent for a variety of different environment specific side effect metrics without major modification to the overall structure of the method.

In addition to training both $A_\theta$ and $Z_\psi$ from scratch as shown in Algorithm 1, we also perform zero-shot generalization of a previously trained $Z_\psi$ from a different environment to investigate whether the concept of side effects can be abstracted out of the environmental dynamics and the intricacies of the task. In this case, lines 11-15 from Algorithm 1 are not performed as no updates for $Z_\psi$ are required, with $Z_\psi$ only being used to modulate the behavior of $A_\theta$ via the distance metric regularization.

### 3.4 Tracking the Champion Policy

The SafeLife suite includes a complex set of procedurally generated environments, which can lead to a significant amount of variability throughout training and testing episodes. In order to account for this variability, we track the best policy throughout the training process for a fixed set of test levels for the different metrics we care about, specifically episode length, performance ratio and side effects, as described in Algorithm 2.

The champion policies operate on test levels, where no learning occurs, and track test-level metrics. This is particularly relevant to the side effect metric, described in Section 2, where we track the episodic side effect even though training occurs with frame-by-frame impact measure. We describe these metrics in greater detail in Section 4.

---

**Algorithm 2** Champion Policy Tracking

---

1: Initialize Training and Champion Policy $C_\theta$
2: **for** Every $k$ Environment Steps **do**
3:      Evaluate task agent $A_\theta^k$ on fixed set of test levels
4:      **if** $Score_{A_\theta^k} > Score_{C_\theta}$ **then**
5:          $C_\theta = A_\theta^k$
6:      **end if**
7: **end for**

---

## 4 EXPERIMENTS

Our experiments contain the following algorithmic runs, where we log the results for $A_\theta$:

- The reward-penalty baseline method described in (Wainwright and Eckersley, 2020) where the impact penalty of a given action is subtracted from the reward the agent receives for that particular frame.

- SARL agents where both the actor $A_\theta$ and virtual safety agent $Z_\psi$ are training from scratch on the same environment using the Jensen-Shannon Distance as well as the dual formulation of the Wasserstein Distance described in (Pacchiano et al., 2019)

- SARL agents where $A_\theta$ is trained while $Z_\psi$ is taken zero-shot from a previous training run on a different environment. The main purpose of this experiment is to show that the concept of side-effects in the SafeLife suite can be abstracted from the specific task (prune vs append) and the specific environment setting (still vs dynamic). The ability to extract a notion of side effects that does not rely on environmental signal for every frame enables us to train the virtual safety agent $Z_\psi$ only once, usually on the simplest task, which can then be used to influence any agent on any subsequent task.

We conduct our experiments on four different tasks in the SafeLife suite: prune-still, append-still, prune-dynamic, append-dynamic. As described in Section 2, dynamic environments have natural variation independent of the actions of the agents, while all changes in still environments can be traced back to the actions of the agent. We evaluate our champion policies $C_\theta$ for the task agent every 100,000 environment steps on the episode length across a set of 100 different testing environments whose configurations are not part of the configurations of environments used in the training process. The length of an episode is the number of steps the agent takes to complete an episode, where a shorter length indicates that the agent can solve the task better and more efficiently. In our results in Section 5 we show the standard error of the champion measured in performance given by the ratio $\frac{\text{agent reward}}{\text{possible reward}}$ and the cumulative side effect measure described in Section 2 and Wainwright and Eckersley (2020).

The experimental results have a strong dependency on hyperparameters chosen, specifically the impact penalty fraction in the reward penalty baseline baseline and the regularization parameter $\beta$ in SARL. Changing these parameters generally results in non-linear trade-offs between episode length, performance and side effects, meaning that policies with high performance often have high side effects and policies with low side effects often have low performance. In the cases of low side effects and low performance, the agent does not perform any significant actions that would either negatively (side effect) or positively (reward) disturb the environment. Our ideal goal is to have a policy that is both performant on the task and has low side effects. As such, in Section 5 we describe results of experiments that in our best judgement represent the best cases of such policies, and apply the same regularization hyperparameters across all environments. The full set of our algorithmic and regularization hyperparameters are shown in Appendix A. As discussed in more detail in Section 6, for future work we aim, and encourage others, to obtain Pareto optimal frontiers that describe the trade-off for the regularization hyperparameters more thoroughly.

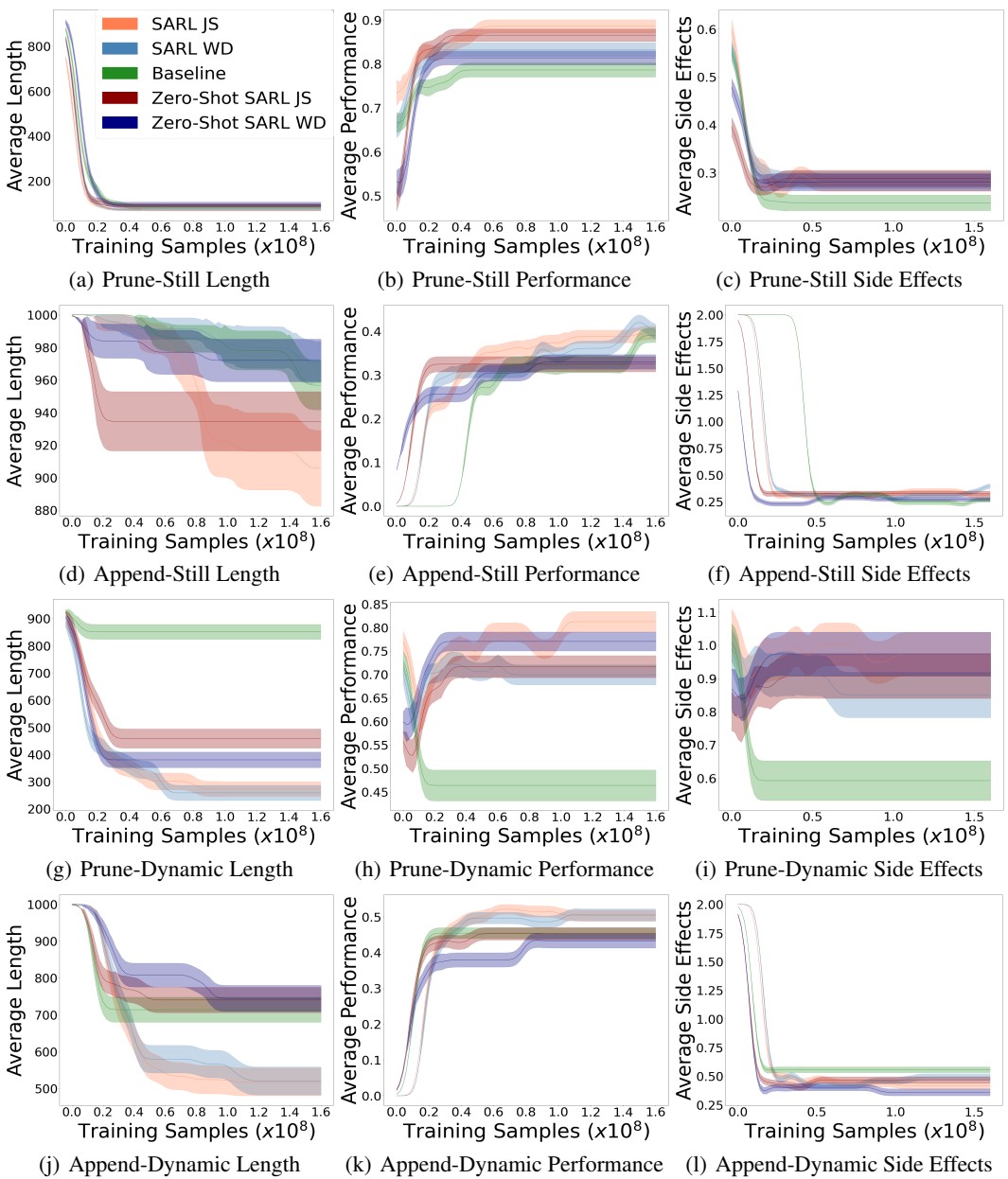

Figure 3: Length Champion for SafeLife Suite of *1. prune-still (a-c), 2. append-still (d-f), 3. prune-dynamic (g-i), 4. append-dynamic (j-l)* tasks evaluated for 100 testing environments every 100,000 steps on *Episode Length (left column)* where shorter is better, *Performance Ratio (middle column)* where higher is better, and *Episodic Side Effect (right column)* where lower is better

# 5 RESULTS

The results of the experiments shown in Figure 3 demonstrate that a virtual safety agent trained on one task in the SARL framework can generalize zero-shot to other tasks and environment settings in the SafeLife suite, while maintaining competitive task and side effect scores compared to the baseline method. This allows us to abstract the notion of safety away from the environment specific side effect metric, and also increase the overall sample efficiency of the SARL method for subsequent training runs. The SARL methods that are trained from scratch also show competitive task and side effect scores compared to the baseline method.

In the still environment we chose to generalize the virtual safety agents from *prune-still* and *append-still* to the other tasks using both distance metrics. The results show that zero-shot generalization of $Z_\psi$ matches the behavior of SARL trained from scratch, as well as matching or outperforming the baseline method on episode length and performance.

**Prune-Still Environment (Top Horizontal):** The reward penalty baseline matches the episode length of all other methods while maintaining a slightly lower performance and side effects. All SARL methods, including both metrics and zero-shot SARL, generally perform equally well on length and side effects while the $SARL_{D_{JS}}$ has a slightly better performance than $SARL_{D_{WD}}$.

**Append-Still Environment (2nd Horizontal):** The reward penalty baseline generally matches the performance and side effects of the SARL methods, while slightly underperforming $SARL_{D_{JS}}$ on episode length. $SARL_{D_{JS}}$ generally performs better on episode length compared to the other methods, both in training from scratch and the zero-shot experiments.

For the dynamic environment we chose to generalize the virtual safety agents trained on *prune-still* and *append-still* to the dynamic environment tasks using both distance metrics. In the zero-shot experiments, we applied the version of $Z_\psi$ that is furthest away from the given setting, meaning *prune-still* is generalized to *append-dynamic* and *append-still* is generalized to *prune-dynamic*. The results show that zero-shot generalization of $Z_\psi$ matches the behavior of SARL trained from scratch, as well as matching or outperforming the baseline method on some metrics.

**Prune-Dynamic Environment (3rd Horizontal):** In this environment, we observe that the baseline method cannot solve the task, as shown by the fact that the episode length does not decrease significantly. However, it incurs very little side-effect cost. This indicates that the baseline agent is acting safer by not doing much in the environment, but actually fails to solve the primary task. All SARL agents outperform the baseline on episode length and performance ratio, indicating that SARL effectively learns the task.

**Append-Dynamic Environment (Bottom Horizontal):** The reward penalty baseline generally matches the behavior of the zero-shot SARL methods on episode length, performance and side effects. The SARL method trained from scratch, both $SARL_{D_{JS}}$ and $SARL_{D_{WD}}$ outperform the baseline as well as the zero-shot experiments on episode length and slightly on performance

## 6 DISCUSSION

In this work, we explored the prospect of regularizing the loss function of an RL agent using distance metrics that encapsulate a notion of safe behavior. We believe this work shows a promising approach to train RL agents in environments where side effects are important. Our method allows a safe policy to be trained once and transferred across tasks within the same environment. This may be especially important when the safety metric itself is difficult to calculate – for example, if it requires human feedback. As mentioned in Section 1, side effects are often difficult to define, especially when interwoven with the primary task. As such measuring and interpreting side effects is an ongoing area of research, which is why we designed SARL to be flexible to various side effect metrics.

The idea of using suitable distance metrics to co-train multiple RL agents has a variety of future research directions. One such avenue is the development of new distance metrics, including different variations of the Wasserstein Distance, as well as ones that can exploit various channels of information that we did not consider in out work (Parker-Holder et al., 2020). The ideal distance metric would capture both the information richness from the different channels and encode a notion of a safety objective which can then be transferred to the primary agent to influence its behavior.

There also exists a great opportunity to apply techniques from multi-objective optimization to side effect problems. The literature is rich with multi-objective optimization problems in supervised learning (Ma et al., 2020) (Sener, 2018) and reinforcement learning (Yang et al., 2019) (Xu et al., 2020) that show promising approaches to adopt a robust multi-objective framework to the side effect problem. The greatest promise of a multi-objective framework is the possibility of obtaining Pareto fronts (Yang et al., 2019) describing the optimal trade-off between task performance and safety in a given environment, which would be immensely valuable to making decisions in real-world environments.

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

## A    IMPLEMENTATION DETAILS

| Hyperparameter | Value |
|---|---|
| $\gamma$ | 0.97 |
| Learning Rate | $3e^{-4}$ |
| Batch Size | 64 |
| Epochs per Training Batch | 3 |
| Environment Steps per Training Iteration | 20 |
| PPO Entropy Weight | 0.01 |
| PPO Entropy Clip | 1.0 |
| PPO Value Loss Coefficient | 0.5 |
| PPO Value Loss Clip | 0.2 |
| PPO Policy Loss Clip | 0.2 |

Table 1: Hyperparameters for PPO-SARL

| Environment | Baseline | SARL JS | SARL WD | SARL Zero-Shot JS | SARL Zero-Shot WD |
|---|---|---|---|---|---|
| Prune-Still | 0.3 | 0.01 | 0.01 | 0.005 | 0.005 |
| Append-Still | 0.3 | 0.01 | 0.01 | 0.005 | 0.005 |
| Prune-Dynamic | 0.3 | 0.01 | 0.01 | 0.005 | 0.005 |
| Append-Dynamic | 0.3 | 0.01 | 0.01 | 0.005 | 0.005 |

Table 2: Hyperparameters for Side Effect Procedures – Baseline: Impact Penalty; SARL: $\beta$

## B    SAFELIFE TASKS

In this paper, we tackle four different tasks in the SafeLife environment:

- *Prune-Still*: In this task, the agent removes red cells placed at different locations throughout the environment. Given the still nature of the environment, there are no variations occurring in the environment other than the ones caused by the agent's actions.

- *Append-Still*: In this task, the agent fills in patterns specified by a blue background placed at different locations throughout the environment. Given the still nature of the environment, there are no variations occurring in the environment other than the ones caused by the agent's actions.

- *Prune-Dynamic*: In this task, the agent removes red cells placed at different locations throughout the environment. The dynamic nature of the environment, where cell patterns oscillate in certain areas, creates additional complexity in the types of cell patterns that the agent interacts with. These types of dynamic variation make completing the task more difficult for the agent, as the distinction between variation caused by the agent's actions and environmentally driven variation is often inherently linked.

- *Append-Dynamic*: In this task, the agent fills in pattern specified by blue placed at different locations throughout the environment. The dynamic nature of the environment, where cell patterns oscillate in certain areas, creates additional complexity in the types of cell patterns that the agent interacts with. These types of dynamic variation make completing the task more difficult for the agent, as the distinction between variation caused by the agent's actions and environmentally driven variation is often inherently linked.

## C    LIMITATIONS AND FUTURE DIRECTIONS OF SARL

SARL relies on training two different agents and taking a probabilistic distance between their policies to regularize the loss of the primary task agent. As discussed in section 1, finding good side effect metrics is a difficult task. Moreover, a side effect metric that is effective for the penalty method described in Wainwright and Eckersley (2020), might not be ideal for SARL safety agent training. As results from this paper suggest, training the safety agent on that metric often encourages the safety agent to stand still for the entire episode, because the safest action is often no action at all. The limitations related to having rich metrics to train both agents also apply outside of SafeLife, where a designer would have to create appropriate functions for the task and safety objectives, and test their effectiveness and generalizability across various situations. Our work suggests that safety agents trained on the SafeLife side effect metric can generalize zero-shot within SARL to other environment and task settings in SafeLife. However, it is unclear whether policies trained in SafeLife could generalize to different environments outside of SafeLife, as generalization of RL policies remains a major area of research.

The above limitations create opportunities for various extensions on SARL. Given that the safety agent of SARL is flexible enough to being trained with any "safety reward", we think that a valuable extension for future work could be to explore how various agents trained on a diversity of metrics, such as variations of the SafeLife metric and metrics in the literature (Turner et al., 2020), perform on their own and within the SARL framework. Safety agents could also be trained on variations of intrinsic reward metrics (Pathak et al., 2017; Eysenbach et al., 2018) that are modified for a side effect scenario. Furthermore, the current version of SARL only applied distances between the policy outputs of $A_\phi$ and $Z_\psi$, but there are many more ways to characterize an agent, such as state- or reward-based trajectories, that might require different distance formulations. As an extension of SARL, one could research the effects of different methods of calculating distances of various policy characteristics to obtain desired behaviors.

## D    DESCRIPTION OF RESULT FIGURES

The figures in Figure 3 show three different metrics that provide insight into how an agent performs in SafeLife.

**Episode Length:** The left vertical pane shows the length of an episode. Since the agent is trying to complete a task and reach the level exit in as few steps as possible, a shorter episode length is generally better. In the top left image we can see that for prune-still, the easiest task, all training methods perform equally on the episode length. On the other tasks shown in the left vertical pane, the various methods show different behaviors for episode length.

**Performance:** The middle vertical pane describes the reward fraction described in Section 4, which shows the fraction of all possible rewards the agent was able to achieve within an episode. Since the agent requires a minimum reward to be able to reach the level exit, while also being incentivized to reach the exit as soon as possible, this serves as a surrogate metric to see how quickly the agent learns the task. As expected, we can see that this metric is highly correlated with the episode length and that all methods generally perform similarly on this metric when they are able to solve the task. The only exception here is the baseline in prune-dynamic in the third horizontal, where the baseline is not able to learn the task effectively.

**Side Effects:** The right vertical pane shows the side effect results based on the episodic metric described in Section 2. We aim to minimize this metric while maintaining strong performance on the other two. In our results, we see that in prune-dynamic (3rd horizontal) the baseline agent achieves a better score on side effects but also does not solve the task, which is not desirable. In append-still (2nd horizontal) all agent show mostly similar behavior; in prune-still (top horizontal) the baseline agent does slightly better on side effect but also achieves a slightly worse reward fraction, and in append-dynamic (bottom horizontal) the SARL methods achieve slightly better side effect scores.

