# OpenReview forum: "Safety Aware Reinforcement Learning (SARL)"
_ICLR.cc/2021/Conference — Reject_

### Official Review · AnonReviewer1 · 2020-10-25
**Although an important direction and promising progress, I have concerns about the method and the results**

**Rating:** 4
**Confidence:** 4

**Review:**

This paper proposes a safety-aware reinforcement learning algorithm that learns to perform tasks with minimal side-effects. The key idea is that a safety policy is learned independent of the task reward. When learning the task, this safety policy is incorporated by minimizing the distance between the task agent and the safety agent. In this way, the paper claims that the safety agent can be generalized to different tasks. The method is tested on SafeLife Suite, and its performance can match task-specific safe learning baselines.

Safe reinforcement learning is an extremely important research area, when we need to apply reinforcement learning to real-world applications, such as robotics, recommendation system, power grid, etc. This paper works in this direction and addresses the key challenges, including how to learn generalizable safety agents. While I think that the paper is promising, I have the following three major concerns:

1) The "side effect metric" is not clear to me. The description in Section 2.1 is high-level and vague. More rigorous mathematical definition is preferred here. For a safe learning paper, it is extremely important to clearly define what safety means. Is the "side effect metric" the same as the "safety metric" in Line 12 of Algorithm 1? Reading from the text, it seems that the side effect metric is calculated per episode, while the safety metric is per step.

2) Section 3.4 seems to leak the testing set into training. One claim of this paper is that the learned safety agent is generalizable: Z(\psi) can be taken zero-shot from previous trained environments. However, during training, by tracking the Champion policy , decisions are made based on the performance on the testing environments, by retaining the policy that performs the best in the testing environments. If my understanding is right, this makes any claim about generalization less convincing because the training directly optimizes the policy in the testing environments.

3) Intuitively, I do not understand how Algorithm 1 could work. According to Algorithm 1, the training of the safety agent Z(\psi) is totally independent of the task, whose only objective is to be safe. If it is the case, the learned safety agent would not move or take action at all. The action distribution P_Z_\psi would be concentrated on the zero action. This would make optimizing A(\theta) using the loss (eq. (1)) extremely difficult. This might explain why the paper observes that the hyperparameter \beta is difficult to tune.

Here are some minor suggestions about writing:
1) A brief description of the 4 tasks is needed (prune-still, prune-dynamic, append-still, append-dynamic) to make this paper more self-contained. If the page-limit is a concern, this description could be added to the Appendix. Otherwise, it is difficult for readers to understand the difficulties and the usefulness of these tasks.

2) "Line 13-17 from Algorithm 1": The pseudo-code ends at Line 16.

For the above reasons, I would not recommend acceptance at this time.

---

> ### Author Response · Authors · 2020-11-11
> **Discussion of Reviewer Comments**
>
> Thank you for your review and comments. We would like to address the concerns you identified directly:
> 1. We have updated our description of the side effect metric in Section 2 of our new draft. There we describe the difference between the frame-by-frame metric used for training the SARL agent and the episodic metric we used to report our results in Section 5.
> 2. The training and testing sets refer to different environment configurations within a given task. Let’s take prune-still as an example: Within prune-still there is a large set of different environment configurations that lead to different maps in SafeLife for which the agent will attempt to solve the prune-still task. Our process takes out a test set of configurations that are not used during training and tracks the champion policy (of the primary task agent) within them purely to report test performance. The safety agent never interacts with the test environments in any of the training runs, and the generalization claim pertains to transferring the safety agent across different environments, meaning that a safety agent trained on prune-still will be applied in append-dynamic, etc. We hope this clarifies the process.
> 3. Your intuition that the safety agent will converge towards a “no action” policy is what we often observed. We also believe this a major reason why finding a good \beta is difficult to achieve. The “no action” policy, however, does not apply to all states that the task actor visits due to the complexity of the SafeLife environment. One potential mitigation for this problem is to modify the reward signal the safety agent is trained on, such as incentivizing it to reach only the level exit as safely as possible. We would also like to add that since task performance and safety often come at a trade-off, it is difficult to define an optimal solution. As we discuss in Section 6 and in our general comments, we plan to extend this work in the future to cast it as a multi-objective RL problem where such trade-offs can be better analyzed.
> 4. We have added more detailed descriptions about the different tasks. We have currently placed them in the appendix due to space concerns.
> 5. Thanks for pointing out this typo.

---

> > ### Comment · AnonReviewer1 · 2020-11-25
> > **Thanks for your response.**
> >
> > You response clarified the definition of side effect metric, which is very helpful. However, I am still concerned about the fact that the safety agent does not care about the task, or the task agent does not affect the update of the safety agent. This makes the training less robust and may need excessive tuning to make it work.

---

> > > ### Author Response · Authors · 2020-11-25
> > > **Task agnostic abstraction of safety is a key feature**
> > >
> > > Thank you for your comment.
> > >
> > > The primary goal of our work is, in fact, to abstract a notion of safety independent of a given task. A key motivation of this goal comes from human notions of safety - which are often pre-formed and often task agnostic.
> > >
> > > For example, we understand that colliding with a fragile object is generally unsafe. Thus, if we need to learn how to navigate in a complex environment that we have never encountered before, we would generally learn to do so in a way that avoids collisions with fragile objects - even if we have never seen those specific objects before. While we will still need to find an optimal policy to solve the primary task, we will generally not need to retrain our understanding of safety. Rather, our pre-formed notions of safety would modulate our own actions as we explore and train ourselves to discover performant strategies.
> > >
> > > The safe agent in our work embodies that notion of task agnostic safety. Similar to the human scenario above, the safe agent modulates the policy of the primary agent as it trains on new tasks.
> > >
> > > We investigate the performance of such "zero-shot" safe agents where the safe agents are ported zero shot to tasks that they have never seen before, which we compare with safe agents that are trained from scratch for those same tasks - this configuration would be the equivalent of an agent that is trained specifically for a given task.
> > >
> > > Our results in Fig 3 show that the zero shot performance is similarly performant as the setting where the safe agent and main agent are jointly trained on the same task. You are correct that there is some tuning required - as we describe in Section 4. However, this formulation is relatively simple - with only the regularization parameter (beta) needing any significant tuning, as is common in many regularization formulations.

---

### Official Review · AnonReviewer4 · 2020-10-28
**Interesting work toward reducing the unwanted side effects of the actions of a reward-maximizing reinforcement-learning agent.**

**Rating:** 6
**Confidence:** 4

**Review:**

The paper aims to reduce the unwanted side effects of the actions of a reward-maximizing reinforcement-learning (RL) agent. The authors study a framework in which the environment issues a metric that measures the total side effects of the agent's actions at the end of each episode. The work's proposed solution trains an agent who focuses on the total reward and another agent who minimizes the total side effects. The authors then empirically investigate the effectiveness of combining the two agents via a distance measure between the policies that unifies them into one.

I find the following items strong points in the submission:
* The posed problem is relevant in the context of safety in AI.
* The chosen testbed for the experiments matches the goals and premises of the posed problem.
* The empirical results suggest that the proposed method is effective.
* The discussion regarding the choice of the distance measure is thorough and makes sense.

On the other hand, I find the following issues as weaknesses in the submitted manuscript:
* There are no theoretical developments to demonstrate whether the reported results generalize beyond the adopted environment settings and the value assigned to the parameter beta or not.
* The paper dives right into introducing the loss function in Section 3 without establishing the required notation and preliminary materials. A brief summary of the task agent and the virtual safe agent descriptions is currently provided in the caption of Figure 2. In my opinion, Section 3 would read better if the authors append a preliminaries section wherein they establish the notations and the descriptions of the task agent and the virtual safe agent.
* The loss function adopted in equation (3) provides little room for theoretical developments. The original paper that introduces the PPO algorithm (Schulman et al., 2017) offers multiple loss function choices. In my opinion, the combination of the Jenson-Shannon distance with the loss function that incorporates the KL divergences enables the authors to study their proposed algorithm beyond empirical results.

I find the posed problem relevant in the context of safety in AI and the suggested method well-motivated and intuitive. I believe the submission is far from theoretically solving the posed problem; however, the methodology alongside the promising empirical results that the manuscript offers may be of interest.

---

> ### Author Response · Authors · 2020-11-11
> **Discussion of Reviewer Comments**
>
> Thank you for your review and comments. We would like to address the weaknesses you identified:
> 1. It is true that we primarily focused our work on the SafeLife suite and cannot make any statements beyond the environments in SafeLife. We believe, however, that SafeLife provides a rich set of different settings that allow us to demonstrate the different notions we discuss in the paper. We believe that a thorough theoretical framework for generalization outside of the SafeLife suite is beyond the scope of the paper. A theoretical framework for generalization of RL agents, as well generalization neural networks overall, are broader fields of research and open problems in the deep learning community.
> 2. Thank you for this feedback. We have updated our notation and further descriptions in our new draft.
> 3. Our current framework assumes that the task agent employs a loss function given from an established RL algorithm. Based on your feedback, we will add a deeper discussion on further development of different loss functions that involve probabilistic distances. We believe conducting this study would not be feasible within the scope of this paper, but we agree that a more thorough theoretical discussion is important.

---

### Official Review · AnonReviewer3 · 2020-10-28
**Timely and important topic, addressed with a simple but neat idea - results are a bit hard to put into perspective / generalize.**

**Rating:** 6
**Confidence:** 4

**Review:**

**Update after authors' response**
I want to thank the authors for their responses. My responses to the authors' comments are in the respective threads.
---

**Summary**
The paper addresses the timely and important problem of how to train RL agents such that they solve desired tasks while not engaging in undesired behavior that is not explicitly specified via the reward function. In particular, the paper focuses on training agents that learn to avoid unnecessary side effects, that is (irreversible) alterations to the environment which are not necessary to solve the task at hand. Experiments are performed on SafeLife, which provides a suite of tasks in an environment (potentially with rich intrinsic dynamics), along with a quantitative measure of the strength of undesired side effects. The main idea of the paper is to co-train an RL policy on this side effect measure with the aim of minimizing side effects. This policy is used for regularizing the reward-optimizing agent during training, such that the trained agent learns to bias its actions towards avoiding side effects when the task allows for multiple viable actions. The paper compares against a strong, previously reported baseline, both in static and dynamic SafeLife environments/tasks. Additionally, the generalization of the side-effect-avoiding policy is tested, by using it for training a reward-optimizing agent on task-versions that the side-effect-avoiding policy has not been trained on.

---
**Contributions, Novelty, Impact**

1) Incorporation of the two (sometimes conflicting) objectives of maximizing reward and avoiding side-effects into a single training objective, where purely reward maximizing actions are regularized by action-distributions from a side-effect-minimizing policy. This is an interesting idea that turns trading off avoiding side effects against reward maximization into a learning problem. I think this is a promising way forward. What I’d like to see in the paper for even greater impact is a clear discussion of the requirements (the objective of avoiding side-effects must be specified as a trajectory-dependent, quantitative function, similar to a reward function), and the current limitations (unclear how to assess “how much” of the task-relevant state-space is well covered by the side-effect-avoiding policy, particularly in the zero-shot setting).

2) Experimental evaluation of the proposed method on *the* state-of-the-art benchmark suite, and comparison against a strong, previously proposed baseline. The results are promising, though it’s hard to distill a very clear message in favor of the method from the results shown. I personally think that’s fine (and to some degree expected when discussing solutions that solve a particular trade-off in a different fashion), but I’d like to see even more of a multi-faceted evaluation and discussion in the paper.

3) The idea of learning a side-effect-avoiding regularizer that generalizes well, e.g. to different tasks under the same environment dynamics. This is very interesting and a promising step towards tackling the side effects problem at scale. It is very nice to see the zero-shot results. To make the paper even stronger and more impactful it would be nice to evaluate the generalization of the trained side-effect-avoiding police in more detail.

---
**Score and reasons for score**
I am (currently) in favor of accepting the paper, though I think that some additional work could improve the strength and potential impact of the work. The topic addressed is timely and very important, and the approach taken is interesting and sensible. Results look promising, and the paper does a great job at presenting the work. To further strengthen the paper it would be nice to discuss results in more detail and potentially perform additional experiments to highlight certain aspects that are “buried” in the current results. Additionally it would be good to say something more substantial about the generalization properties of the side-effect-avoiding policy. While the latter two issues are probably beyond what’s easily possible in the rebuttal phase, I want to strongly encourage the authors to add a short paragraph that clearly states the assumptions/requirements (the strongest assumption is perhaps the presence of a quantitative side effect measure which can be used directly as a reinforcement signal), and current limitations. I am looking forward to the other reviews and authors’ response, and will update my final verdict accordingly.

---
**Strengths**
1) Empirically promising results on a timely and important problem, including the comparison against a strong baseline method.

2) Evaluation of proposed method by: (i) multiple runs to assess statistical significance, (ii) ablation studies regarding the “distance” metric used by the method, (iii) control-experiments regarding the (zero-shot) generalization performance of the side-effect-avoiding policy.

3) Well written paper, with good introduction to the problem and discussion of related literature (given the limited space of a conference-format publication).

---
**Weaknesses**
1) The experimental results shown are interesting and promising, but it’s hard to distill a clear message from the results other than: “the proposed method seems to work on par with a previously proposed method but often makes the trade-offs (between high reward and low side-effects) differently, which makes comparison more difficult”. Drilling down on some of the findings and trying to control for more factors to get a clearer picture would strengthen the results.

2) The generalization of the side-effect-avoiding policy is a very interesting aspect of the work, however the current analysis of how well that generalization behaves is a bit crude. It is unclear to which degree the previously trained side-effect-avoiding policy in the zero-shot regime covers the state-space encountered when solving a particular task. It is also unclear whether the side-effect-avoiding policy in the generalization setting “behaves mostly well overall” or whether it has some severe and potentially even systematic shortcomings (leading to undesired policies) in particular situations of the generalization regime. Addressing this in full generality is of course beyond the scope of this paper, but some more analysis into this issue would be very nice to see (e.g. comparing the zero-shot vs the trained side-effect-policies in isolation, and potentially drilling in on some of the differences encountered).

---
**Correctness**
The construction of the algorithm and training scheme presented in the paper seems correct to me.

---
**Clarity**
The paper is mostly well written, and the method is clearly described. Perhaps two things to improve: (i) the discussion of results could be expanded a bit more, there’s a lot going on in the plot and unfortunately there’s no intuitive message that one can easily take away visually. (ii) To facilitate the flow of the manuscript to readers unfamiliar with SafeLife it would be nice to include a short section describing the side-effect penalty.

---
**Improvements / major issues**
1) The results currently shown are interesting but it’s hard to distill a clear message (which is understandable to some degree, as the paper also points out, because different solutions to a multi-objective optimization cannot be easily compared). It might be worthwhile though to expand the discussion (and perhaps even presentation) of the results a bit more.

2) One of the most interesting aspects of the work is the potential to train a task-agnostic side-effects-avoiding policy that generalizes to a broad range of tasks. The paper demonstrates that this works by applying said policy in a zero-shot setting and analyzing the resulting policy. It would be nice to also do some more comparison of the side-effects-avoiding policies directly (e.g. what is the side effect score when directly comparing a zero-shot Z vs a Z trained on the current task/environment - are there any systematic deviations between the two, do certain biases get baked into the zero-shot Z that can be explained by the tasks/environment-variants it’s been trained on).

3) A clear discussion of the requirements (the objective of avoiding side-effects must be specified as a trajectory-dependent, quantitative function, similar to a reward function), and the current limitations (unclear how to assess “how much” of the task-relevant state-space is well covered by the side-effect-avoiding policy, particularly in the zero-shot setting).

4) Please clarify: why are there separate zero-shot agents shown in prune-still and append-still - shouldn’t they be the same SARL JS/WD since the zero-shot agents have been trained on these two tasks respectively?

5) Please clarify and potentially discuss in the paper: perhaps the main requirement for the method is having a side-effect-strength signal s. This signal must be suitable for a reinforcement learning algorithm to train a side-effects-minimizing policy Z. But if such a signal is available, why not simply combine it with the task-specific reward function r to create a “safe reward function” to train a reward-optimizing agent that avoids side-effects? Would the solution obtained this way be qualitatively different (in some aspects) compared to the solution obtained by the proposed scheme? It’s fine to simply comment on this - the strongest version would include actual control experiments (but I understand that this might not be easily doable).

6) Please comment and potentially discuss in the paper: What is the advantage of co-training Z with A (lines 11-15 of Algorithm 1)? Why not train Z first (e.g. would that improve training stability)?


---
**Minor comments**

A) Please give a few details for the side effect metric that’s used by the experiment (fine to refer to the SafeLife paper for full details, but the rough idea should be in the paper to improve readability).

B) How exactly is it ensured that Z sees the same parts of the state-space that A does (i.e. how is it ensured that Z “explores” similarly to A, which is solving some tasks)? I assume that the actions actually taken (which lead to a certain state on which A and Z are evaluated in line 5 and 6 in Algo 1) are driven by A?

C) P4: “In this formulation, policy characteristics are converted to distributions in a latent space of behavioral embeddings on which the Wasserstein Distance is then computed.”. I have a hard time following this sentence, please consider unpacking it a bit.

---

> ### Author Response · Authors · 2020-11-11
> **Discussion of Reviewer Comments**
>
> Thank you for your very thorough review, we hope that we can clarify some of your concerns through these discussions. In this comment, we would like to focus on the Improvements you outlined and address them directly:
> 1. We agree that a clearer discussion on the key takeaways would be useful. Our primary goal was to train a portable safe-agent that could generalize across multiple tasks that share common dynamics. Our results on zero shot transfer show that the transferred safe agents performed similarly compared to training SARL agents from scratch on the same environment. The primary distinctions from prior work are:
> a. the use of a safe-agent to abstract the concept of safety.
> b. the portability of the safe agent across multiple tasks.
> c. using probabilistic distances to capture differences between policies with different objectives on the same environment
> We did not cast this problem as a multi-objective optimization problem in this paper - however, that is a natural next step for this line of work in the future. As mentioned in our general comment, proper reframing of the framework as a multi-objective setup requires significant future work, which we believe this is out of scope for this paper.
> 2. Based on our understanding of your comment, we already performed those experiments. We call them “from scratch training” and show them in Section 5 in the lighter shade of color. Here, the task agent and safety are trained concurrently on the same environment, and we observe similar behavior to using safe policies that are zero-shot generalized from other environments. Does this address the scenario you are suggesting?
> 3. We have updated the details pertaining to the side effect metric in Section 2 of our new draft. We take the side effect metric from SafeLife as is and use it to train our safety agent. We would like to highlight that we designed SARL to be agnostic to the safety metric used, as the distance function formulation allows for that flexibility.
> 4. In the prune-still environment we take the safety agent trained on the append-dynamic environment, and in the append-still case we take the safety agent trained on the prune-dynamic environment. As we discuss in Section 5, we wanted to take the safety agent trained on the environment that is most dissimilar to the task environment since we believe that would be the most difficult to generalize from. It is not strictly necessary to do that, as we could have one zero-shot agent that is generalized to all environments except for the one it is trained on.
> 5. We already train a baseline policy on a shaped reward that combines the primary objective and side-effect penalty. Since this was the method applied in the original SafeLife paper, we use this method as the “baseline” in our experiments. This method takes the frame-by-frame side effect from SafeLife and subtracts that from the frame-by-frame reward (with a scaling factor). We searched for a good scaling factor as a hyperparameter in our experiments to support the side effect baseline.
> 6. Thank you for this suggestion. Co-training the task and safety agents on the same environment was primarily a practical choice, but we will add experiments that work with a safety agent that is previously trained. Intuitively, we do not expect major changes and agree with you that it should improve training stability.
>
> Regarding the Minor Comments:
> 1. We have added more details on the side effect metric in our new draft.
> 2. The current environment includes a fully visible state space; we can therefore pass the entire state to A and Z when we train A. In the current setup (using an on policy method like PPO), it is difficult to ensure exploration of the same states. If we were to use an off-policy algorithm, such as DQN, we can use the replay buffer to ensure that A and Z are drawing from the same distribution of states during their training.
> 3. This sentence refers to how the Wasserstein distance is computed using the method described in Pacciano et al. They construct a space of test functions in a latent behavioral space and compute in that space. Essentially what happens is: 1. A trajectory (defined by the user) is given 2. A function transforms that trajectory to latent embedding space 3. The WD distance is computed in that space iteratively via test functions. The original paper provides a lot more detail on this process, which essentially allows one to take any definition of a trajectory and compute a distance on it.
> Please let us know if we addressed your points and if you want to continue to discuss more.

---

> > ### Comment · AnonReviewer3 · 2020-11-17
> > **Thank you for the detailed commets!**
> >
> > Let me clarify some of the issues raised.
> >
> > 1. Yes, I understand the broader goal. What I meant was to expand the actual description/discussion of the results in the paper to literally walk the reader through the most important parts to pay attention to in Fig. 3. The figure has a lot of information that cannot easily be grasped. A bit more guidance for the reader would be nice to have.
> >
> > 2. That's not quite what I had in mind (I think). What I meant was to directly report the side-effect score (r_{side-effect}) of Z (without any involvement of A). As far as I understand Sec. 5 shows results for evaluating r_{side-effect} for an agent A regularized via Z (either a co-trained Z or a Z trained on some other task). The reason for asking for such a comparison is that it might give some insight into the generalization performance (and gaps!) of Z in the most direct way possible (not indirect evaluation by studying A trained via various Z). Please let me know if these results are actually shown in Fig. 3 (which panel, which lines?) if I'm still misunderstanding Fig 3.
> >
> > 3. What I meant here was adding a 'limitations' section to the paper, where the assumptions/requirements and limitations are clearly and concisely stated. To me, the biggest limitation of applying the work outside of SafeLife is the requirement of having a rich side-effect metric (rich enough to train an agent on). I would even argue that having such a function in the first place is the main difficulty. This might be obvious when talking about SafeLife, but should be clearly discussed as a problem w.r.t. the applicability of the method in the paper.
> >
> > 4. Thanks!
> >
> > 5. Thanks for clarifying the effort of searching for a good scaling factor to strengthen the baseline. Besides the empirical evaluation my comment was also aimed at the conceptual level: if a rich safety-metric signal for an environment (not only a specific task in that environment) is available, why not use that signal directly as a regularizer-term in the reward function? What are the (theoretical?) advantages of using it to train a safe policy (which is then used as a regularizer)?
> >
> > 6. Thanks for clarifying!

---

> > > ### Author Response · Authors · 2020-11-19
> > > **Response To Revised Comments**
> > >
> > > Thank you for your response. Let’s go through your revised points in more detail:
> > > 1. Thank you for the suggestion. We have added a section to the appendix that describes the figures in the results section in more detail. We plan to add this description, along with the section on limitations, in the final version of the paper.
> > > 2. Thank you for the clarification. You are right in your understanding that the figures only show the results for the task actor A and not the safety actor Z. We will re-run our experiments to include logging  for Z_psi in the test environments in addition to A_phi, and add those figures into the appendix. We hope to be able to include them in the next update of the paper if the experiments finish in time. We will let you know if/when a revision with those figures is uploaded.
> > > 3. We have added a section in the appendix to discuss limitations and assumptions more thoroughly. We plan to move this section to the final 9-page version of the paper if accepted. We agree that SARL in its current form relies heavily on the side effect metric, just like any RL agent is dependent on the reward function used for its training. Moreover, the SafeLife metric while a good fit for the impact penalty method might not be ideal for SARL, which is why we believe that a more thorough study of training with different safety metrics, including unsupervised metrics, would be a valuable extension of this work.
> > > 4. You’re welcome.
> > > 5. Learning a separate safe policy can be much more efficient than learning a combined policy when the safety metric is itself expensive to compute. This would be the case, for example, when safety is learned via human feedback. If the safety policy itself is relatively straightforward, or if the safety policy can be applied to many different tasks, then learning the safety policy by itself can be much more sample efficient in safety metric calculations than an agent that learns via a combination of the safety metric and its standard reward. Moreover, a separate safety policy, once learnt, should be able to distinguish between side effects that are within the agent's control and those that are not. In contrast, the safety metric by itself may introduce considerable noise into the training procedure of the primary task if it includes side effects that are not caused by the agent and are instead part of the inherent environment dynamics. This is particularly important in the dynamic environments within SafeLife. We also want to highlight that the SARL framework allows for more flexibility in how to compute the distance between policies, which gives more options on how to act in complex spaces.
> > > 6. You’re welcome.

---

> > > > ### Comment · AnonReviewer3 · 2020-11-23
> > > > **Thanks for the quick response!**
> > > >
> > > > Happy to see the interactive part of the reviewing process working well via OpenReview, even happier to see the additional writing (1. and 3.) and the new experimental results (2.).
> > > >
> > > > I like the reasoning in 5. Perhaps worth adding (some of) it to the paper.

---

### Official Review · AnonReviewer2 · 2020-10-30
**Unclear problem formulation and not fully warranted algorithm**

**Rating:** 3
**Confidence:** 4

**Review:**

This paper aims to address the issue of mitigating side effects in policy learning. The authors propose an algorithm SARL, which uses a safe policy to define a regularization term for penalizing the agent's actions deviating from the safe agent in policy learning. In the experiments, four variations for SARL are shown and compared with a baseline method based on reward penalty. The proposed algorithm is competitive across the experiments presented in the paper.

I think side effects and safety in reinforcement learning is an important issue. However, this paper does a bad job in describing the problem it wishes to address and, therefore, it's unclear whether the proposed algorithm really achieves that goal.

1. The main motivation of this paper is to mitigate the side effects in learning. However, the definition of side effects were never given. It's only until Algorithm 1 is presented where the paper mentions a safety metric that the safe agent aims to optimize (is this the same s appearing in A(s|theta) and Z(s|psi) in Sec 3.2?), which however is not defined. Therefore, I do not fully understand what the objective of this learning algorithm wants to achieve. From the paper's vague description, it seems like the goal is that the learner should have high performance in the original reward while not causing high side effects. This is a multi-objective MDP problem or at least can be framed as a constrained MDP. However, the proposed algorithm, based on simple regularization with a constant weight, can address neither of these two criteria. I am wondering if the authors consider to more explicitly outline the solution concept they wish to obtain. Current hand-wavy description makes me difficult to judge whether the proposed algorithm actually solves the problem they wish to solve.

2. In Algorithm 1, since the safe agent Z is updated independently of the progress of the learner agent A, when there's only a single environment, there is no point of distinguishing the so-called "zero-shot" and the online version, as in high level this dependency allows us to pretrain the safety agent alone beforehand and get the same results. Or do the authors mean zero-shot in the sense that the safe agent is trained on a different set of environments and the online version means they're trained on the same environment?

3. In the paper, the authors write multiple times that a difficulty in this problem setup is that the side effects are difficulty to define. But it seems that the proposed algorithm assumes some safety metric. How are the two related precisely? And what is that used in the experiments?

4. What is S[\pi_theta] in (3)?

Overall, I think the paper is rather incomplete and therefore I do not recommend acceptance.

---

> ### Author Response · Authors · 2020-11-11
> **Discussion of Reviewer Comments**
>
> Thank you for your review. We would like to address your main concerns directly:
>
> 1.
>
> a. We agree that the description of side-effects was not completely clear in the paper and have updated the description in our new draft as mentioned in our response to all reviewers. For SARL training, we assume that the environment provides a definition of the side effect. As we describe in Section, SafeLife calculates a side effect signal by taking the deviation between a baseline state and the current state. This is the frame-by-frame metric we use to train our agents. We also use the episodic side-effect to report our results where this difference is calculated between a distribution of states at the beginning and end of an episode to account for environment dynamics.
> b. Thank you for pointing out the inconsistent notations. We have updated the paper, including Algorithm 1, to correct this instance as well as a few others we noticed. In the current version, s refers solely to the state of the environment. In the old notation, the ‘s’ in Algorithm 1, line 1-2, ‘s’ referred to the state of the environment and in Algorithm 1, line 12-14 we referred to the side effect metric as ‘s’ in green color.
> c. You are correct that the multi-objective MDP formulation is very applicable here. As we discuss in Section 6, this is a future direction that we are highly interested in since we believe it would be a more effective way to understand the trade-offs between task performance and safe behavior. Our primary goal in this paper was to encapsulate the concept of safety into a safe actor - that could then be used to modulate the behavior of a primary agent on different tasks in the same dynamic environment. This mitigates the need to define or learn a penalty factor for every task that shares similar dynamics.
> 2. In the zero-shot experiment, the safety agent is trained on a different environment and then transferred over to a new environment without re-training. In the case of online co-training, the safety agent is trained in conjunction with the regular task agent on the same environment.
> 3. In this paper, the side-effect metric is defined by the environment, as we previously mentioned in Point #1. Side-effect metric and safety metric are used interchangeably - we have updated the manuscript to keep the terminology consistent. Our discussion in Section 1 and Section 6 focused on the general difficulty in developing an effective side effect metric which continues to be an active area of research.
> 4. S[\pi_theta] in Equation 3 refers to the entropy of the policy as described in the original PPO paper.
>
> Please let us know if we addressed your points and if you want to continue to discuss more.

---

> > ### Comment · AnonReviewer2 · 2020-11-24
> > **Still unclear problem formulation**
> >
> > Thanks for the responses and the updated paper.
> >
> > I'm still unclear about the "precise" goal that this algorithm is trying to achieve. While it's clearer what the side effects are now, I still do not understand what kind of behaviors we wish the agent to achieve, given such a side effect definition. Is there a threshold on how much side effect that the agent can induce? Is the side effect treated as penalty on the rewards? Should the learner reach Pareto front? Why is that solution concept preferred in the application here?  I think the overall issue of this paper is that the desired solution concept is unclear, as I mentioned in the previous comment. The authors propose a method but it is unclear what problem precisely it's trying to solve. I think a more in-depth comparison with relevant solution concepts and literature is mandatory to improve the paper's clarity.
> >
> > Lastly, I think it's misleading to say that the task agent and the safe agent are co-training, because the task agent never affects the update the safe agent in Algorithm 1.

---

> > > ### Author Response · Authors · 2020-11-25
> > > **Response To Revised Comments**
> > >
> > > Thank you for your response. We wanted to reply to some of the questions you raised:
> > > - Precise Goal:
> > > The goal for this paper is a framework that allows us to abstract the notion of side-effects into a safe actor that can then be ported across several tasks and environment settings that share the same underlying dynamics but different primary objectives. In the zero-shot settings, we show that this is possible within the SafeLife suite where a safe agent trained on one task can modulate the behavior of a primary agent on a second task.
> > > - Behavior of the Agent:
> > > In terms of what behavior the agent is trying to achieve, that is a significantly more difficult question because it highly depends on the goal of the practitioner. The ideal policy, of course, would solve the task perfectly without leaving any side effects but that is not possible to achieve with current methods, and often not possible even in principle. As we mention in Section 6, we believe that applying ideas from multi-objective optimization to obtain a Pareto Frontier is a promising idea to further quantify such behaviors to enable practitioners to make better decisions about task and safety performance given their design settings (reward function, side effect metric, algorithm, etc). Given the large scope of such a study, we believe that is a great extension for future work.
> > > - Why is this solution concept preferred?
> > > This solution concept, SARL, is preferred because it is flexible to be trained with different side-effect metrics that can induce different behaviors. As discussed in Section 2,  the role of side-effects of RL policies is an open problem and we anticipate that new formulations of side-effects will be invented in the future and wanted to design a framework that will be flexible to those innovations.
> > > - Co-Training
> > > We agree with your statement that using co-training can be misleading. We will use better terminology in the final paper.

---

### Author Response · Authors · 2020-11-11
**Overall Response to Reviewer Comments**

We thank all reviewers for their comments, suggestions and feedback. As we read through all the reviews, we noticed some common threads that we think are relevant to all reviewers and have motivated us to make updates to our current paper draft, which has been uploaded to replace the previous draft.
1. The first change we made is to standardize the notation used in our method (Section 3) to clear up inconsistencies and to convey the important parts of the SARL framework in a clearer manner.
2. We have also updated and added more detail to the description of the side effect metric used for training the SARL agents in Section 2. These metrics are identical to the ones in the original SafeLife paper, and we agree with the reviewers that more detail on this metric was needed. We hope our changes provide a better description of these metrics and how we applied them in the SARL method.
3. We have also made some changes to clarify the setup of our zero-shot generalization experiments. In each of those experiments we load a safety agent trained on a different environment and apply it to the SARL agent without any updates, only performing lines 1-10 in Algorithm 1. In this set-up, since the safety agent is trained on a different environment than the task agent and does not perform gradient updates, we labeled these experiments as zero-shot generalization.
4. We wanted to also address the comments regarding multi-objective formulation. Proper reframing of the framework as a multi-objective setup requires significant future work, including: 1. choosing preferences functions for the two objectives to cast a proper MOMDP; 2. Develop an algorithm to train an agent on this MOMDP; 3. Develop a method for creating and visualizing Pareto fronts that describe the trade-off between the different objectives for the given method; 4. A study of how flexible the method is to changes in preferences during training and testing. Given the extensive nature of the work required, we believe this is out of scope for this paper.
We have also prepared individual responses to all reviewers and hope to continue this discussion to address any questions that come up throughout this process.

---

### Decision · Program_Chairs · 2021-01-07
**Final Decision**

**Decision:**

Reject

**Comment:**

Based on the paper, reviewers' comments and discussions, and the responses, the meta-reviewer would like to suggest the authors to improve the paper and resubmit.